# Sexual Dimorphism in Body Weight Loss, Improvements in Cardiometabolic Risk Factors and Maintenance of Beneficial Effects 6 Months after a Low-Calorie Diet: Results from the Randomized Controlled DiOGenes Trial

**DOI:** 10.3390/nu13051588

**Published:** 2021-05-10

**Authors:** Inez Trouwborst, Gijs H. Goossens, Arne Astrup, Wim H. M. Saris, Ellen E. Blaak

**Affiliations:** 1Top Institute Food and Nutrition (TIFN), 6708 PW Wageningen, The Netherlands; g.goossens@maastrichtuniversity.nl (G.H.G.); e.blaak@maastrichtuniversity.nl (E.E.B.); 2Department of Human Biology, NUTRIM School of Nutrition and Translational Research in Metabolism, Maastricht University Medical Center+, 6229 ER Maastricht, The Netherlands; w.saris@maastrichtuniversity.nl; 3Department of Nutrition, Exercise and Sports, Faculty of Science, University of Copenhagen, 1165 Copenhagen, Denmark; ast@nexs.ku.dk

**Keywords:** weight loss, sexual dimorphism, low calorie diet, cardiometabolic risk factors, glucose homeostasis

## Abstract

A low-calorie diet (LCD) is an effective strategy to lose weight and improve cardiometabolic risk factors, however, sexual dimorphism may be present. This study aims to investigate sexual dimorphism in cardiometabolic risk factors following weight loss and after weight maintenance. 782 overweight/obese participants (65% women) of the DiOGenes trial followed an 8-week LCD (~800 kcal/day), with a 6-months follow-up weight maintenance period on *ad libitum* diets varying in protein content and glycemic index. Men lost more body weight during the LCD period (−12.8 ± 3.9 vs. −10.1 ± 2.8 kg, respectively, *p* < 0.001), but regained more weight during the follow-up period than women (1.5 ± 5.4 vs. −0.5 ± 5.5 kg, respectively, *p* < 0.001). Even though beneficial LCD-induced changes in cardiometabolic risk factors were found for both sexes, improvements in HOMA-IR, muscle and hepatic insulin sensitivity, triacylglycerol, HDL−, LDL− and total cholesterol, diastolic blood pressure, cholesterol esters, sphingomyelins and adiponectin were more pronounced in men than women (std. ß range: 0.073–0.144, all *q* < 0.05), after adjustment for weight change. During follow-up, women demonstrated a lower rebound in HDL-cholesterol, triacylglycerol and diacylglycerol (std. ß range: 0.114–0.164, all *q* < 0.05), independent of changes in body weight. Overall, we demonstrated sexual dimorphism in LCD-induced changes in body weight and cardiometabolic risk profile, which may be attributed to differences in body fat distribution and metabolic status.

## 1. Introduction

Obesity is as a metabolic disease with major implications for the development and progression of several chronic diseases, including cardiovascular diseases and type 2 diabetes mellitus [1]. With the increasing worldwide prevalence of obesity, interventions targeting obesity and its cardiometabolic complications are increasingly warranted. Weight loss resulting from a low-calorie diet (LCD) is seen as an effective strategy to improve obesity and its related metabolic risk factors [2]. Nevertheless, weight maintenance after successful weight loss is one of the biggest challenges in the management of obesity, while it is of utmost importance to maintain improvements in cardiometabolic risk factors [3]. Interestingly, men and women show distinct responses to a LCD, both in relation to the amount of weight loss, the associated improvements in cardiometabolic risk factors and also how well these improvements are maintained during follow-up [2,4,5].

Firstly, distinct sex steroid hormone concentrations may contribute to sex-specific differences in glucose homeostasis, body composition and energy balance, as evidenced by the deleterious impact of the decreasing estrogen concentrations during menopause on cardiometabolic risk factors [4,6]. Furthermore, women generally have a greater subcutaneous lipid storage capacity as reflected by the larger gluteal-femoral subcutaneous adipose tissue (SAT) depot compared to men [7], which seems protective against the development of several cardiometabolic diseases [8]. In line, differences in plasma lipid(ome) profile are present between overweight and obese men and women [9,10]. In recent years, it has become clearer that these differences may reflect tissue-specific and/or whole-body lipid metabolism [10]. Altogether, these differences appear to be important determinants of sexual dimorphisms in weight loss and its associated metabolic adaptations [5,11,12] and may explain the distinct responses following LCDs that have been reported between men and women [12,13,14]. However, often it is not clear whether sex-specific metabolic responses to LCDs are independent of difference in weight loss or weight regain, or whether differences can be ascribed to sexual dimorphism in baseline (metabolic) characteristics.

A better understanding of the complex interplay between sex and changes in cardiometabolic risk profile following weight loss and during weight maintenance may contribute to the development of more targeted dietary interventions to prevent and treat overweight/obesity and related cardiometabolic complications [4,15]. In the present study, we therefore investigated sex-specific differences in diet-induced weight loss and cardiometabolic risk parameters, including plasma lipidome profile and indicators of insulin sensitivity, in overweight and obese individuals in the large pan-European multicenter DiOGenes study.

## 2. Materials and Methods

### 2.1. Study Design and Population

The DiOGenes study is a large randomized controlled dietary intervention study executed in 8 different European cities. Detailed description of the specific study objective, methods and sample size calculation are described elsewhere [16,17]. Briefly, 938 participants followed an 8-week LCD consisting of ~800 kcal/d (Modifast^®^, Nutrition et Santé, Belgium) to achieve weight loss. When ≥8% of the initial body weight was lost during the weight loss phase, the participant was randomly assigned to follow one of 5 *ad libitum* diets, differing in protein content and glycemic index, for 6 months (n = 555) (Figure 1). The latter period was aimed at weight maintenance. Weight stable adults with a BMI of ≥27 kg/m^2^ and ≤40 kg/m^2^ and <65 years of age and fasting blood glucose concentrations <6.1 mmol/L without history of cardiovascular disease or diseases affecting body weight control were recruited. Local ethics committees approved the study (MEC 05–097), and all participants gave written informed consent before participation in the study. The study was carried out in accordance with the principles of the Declaration of Helsinki. The study is registered under the ClinicalTrials.gov Identifier: NCT00390637.

### 2.2. Clinical Visits

Data of three clinical investigation days (CIDs) were included in this study. CID1 was performed before the start of the 8-week weight loss phase, CID2 after the weight loss phase and CID3 after the 6-month weight maintenance phase. During all CIDs weight and height, waist and hip circumference and blood pressure were determined. Furthermore, after an overnight fast, participants underwent an oral glucose tolerance test (OGTT). Venous blood was sampled before (t = 0 min) and at 30, 60, 90, 120 min after ingestion of the glucose drink. Physical activity was assessed using the validated BAECKE questionnaire, including domains of work, leisure and sports [18].

### 2.3. Biochemical Analysis

Venous plasma and serum samples were stored at −80 °C and analyzed. Both fasting and postprandial glucose and insulin and fasting lipid parameters (free fatty acids (FFA), triacylglycerol (total TAG), HDL-cholesterol, LDL-cholesterol and total cholesterol), C-reactive protein (CRP) and adiponectin were analyzed. Furthermore, plasma lipidome profiles were analyzed (see ‘2.5 Plasma lipidomics’ below for method of analysis). All samples were sent to a central laboratory, depending on the type of analysis.

### 2.4. Insulin Sensitivity

Whole-body insulin sensitivity was estimated using HOMA-IR, which was calculated using the following formula: (glucose_t=0_ (mmol/L) × insulin_t=0_ (mU/L))/22.5. Furthermore, tissue-specific insulin resistance was estimated according to the method of Abdul-Ghani et al. [19], and validated against the hyperinsulinemic euglycemic clamp technique. Previous studies have demonstrated that, using these indices, muscle, liver and adipose tissue insulin resistance are related to distinct cardiometabolic disease risk profiles (for example a distinct lipidome profile) [10,20,21], and have been found to be different in men and women [10,20,22].

Hepatic IR and muscle insulin sensitivity can be estimated based on glucose and insulin values during a 5-point OGTT, using the hepatic insulin resistance index (HIRI) and muscle insulin sensitivity index (MISI), respectively [10,19,20]. HIRI was estimated using the square root of the product of the area under curves (AUCs) for glucose and insulin during the first 30 min of the OGTT using the formula: HIRI = √(glucose_0–30_ [AUC] * insulin_0–30_ [AUC]). MISI was calculated using the following formula: MISI = (dG/dt)/mean insulin during the OGTT. Here, dG/dt is the rate of decay of plasma glucose concentrations during the OGTT, calculated as the slope of the least square fit to the decline in plasma glucose concentration from peak to nadir. MISI and HIRI have been validated against the gold standard hyperinsulinemic-euglycemic clamp, where MISI was validated against the glucose disposal rate and HIRI with the endogenous glucose production [19]. The adipose tissue insulin resistance index (ATIRI) was calculated as fasting insulin (μU/L) × fasting non-esterified fatty acids (μmol/L)/1000.

### 2.5. Plasma Lipidomics

Fasting plasma lipidome analysis was performed for all three clinical visits using liquid chromatography-mass spectrometry as described previously [11]. Briefly, a mixture of internal standards and calibration standards were added to each sample. Then, liquid-liquid extraction was performed using a dichloromethane-methanol (2:1) mixture. Following, lipids were separated on a Ascentis Express C8 2.1 9 150-mm (2.7-μm particle size) column (Sigma-Aldrich, Bellefonte, PA, USA) with the use of an Acquity UPLC system (Waters, Prague, Czech Republic) and visualized and quantified using quadrupole time-of-flight mass spectrometry (Agilent Technologies, Santa Clara, CA, USA) [11]. A total of 140 lipids were detected based on the actual presence of target compounds in a LC-MS raw data file. The target compounds are previously identified by LC-high resolution MS, using the FT-ICR-MS and (relative) retention time [23], resulting in a high degree of certainty [24]

The lipids were grouped into 11 lipid groups species according to Lipid Maps nomenclature (http://www.lipidmaps.org (accessed on 5 April 2020)). The lipid groups are defined as follows: TAG (n = 55), diacylglycerols (DAG; n = 2), cholesterol esters (CholE; n = 3), lysophosphatidylcholine (LPC; n = 11), lysoalkylphosphatidylcholine (LPCO; n = 2), lysophosphatidylethanolamine (LPE; n = 1), phosphatidylcholine (PC; n = 25), alkyl-phosphatidylcholine (PCO; n = 15), phosphatidylethanolamine (PE; n = 3), alkylphosphatidylethanolamine (PEO; n = 3) and sphingomyelins (SM; n = 20). The TAG species was further sub-divided into potential metabolically relevant sub-groups based on composition, saturation and chain length of TAGs as they have been shown to be differentially related to cardio-metabolic risk factors in men and women [10].

### 2.6. Statistical Analysis

In this exploratory analysis, differences in characteristics between men and women were assessed using an independent sample *t*-test. Linear mixed-model analyses were performed to assess the differences in change in cardiometabolic risk parameters between men and women following weight loss (from CID1 to CID2) and after weight maintenance (from CID2 to CID3). The linear mixed-model was performed with cardiometabolic risk parameters as the dependent variable, sex as fixed effect and study center as random effect. Age, body weight at baseline, weight loss and baseline value of the dependent variable were included as covariates in the analyses for the change following weight loss (from CID1 to CID2). For the weight maintenance phase (from CID2 to CID3), age, baseline weight, baseline value of the dependent variable, change of the dependent variable during weight loss, weight change during the weight loss phase and weight maintenance phase, and diet were included as covariates. Since physical activity level was not different between men and women, and was not significantly altered during the intervention, physical activity was not included in the analyses. In case data were not normally distributed, data were Ln-transformed to approximate normality. Z-scores were calculated in order to standardize effect sizes, allowing direct comparison of different variables. To correct for multiple comparisons and limit false positive outcomes, p-values were corrected for False Discovery Rate (FDR) using the Benjamini-Hochberg method. FDR-adjusted *p*-Values (*q*-Values) were used to describe the data with a significance set at *q* ≤ 0.05. Only results of the fully adjusted models are shown (depicted as ‘adj.’). The statistical analysis was performed using the IBM SPSS Statistics software (version 25).

## 3. Results

### 3.1. Clinical Characteristics of the Study Population

782 participants were included in the analysis of the weight loss phase, of which 555 also participated in the weight maintenance phase. Of the total population, 506 (=64.7%) were women. A detailed overview of the clinical characteristics of the men and women in the study are listed in Table 1. Briefly, the average age of men in the study was 42.5 ± 6.0 years (mean ± standard deviation (SD)), compared to 41.0 ± 6.3 years in women (significant different, *p* = 0.001). At baseline, women had lower values for waist-hip ratio, systolic (SBP) and diastolic blood pressure (DBP), fasting glucose, 2-hr glucose, fasting insulin, HOMA-IR, HIRI, total cholesterol, LDL-cholesterol, triglycerides and adiponectin compared to men while HDL-cholesterol, FFA and CRP were higher in women (all *p* < 0.01). BMI, physical activity, 2-hr glucose, MISI and ATIRI, were not significantly different between men and women. The clinical characteristics of the 555 participants participating in the weight maintenance phase are comparable with the complete population characteristics (data not shown).

### 3.2. Weight Change Following LCD

On average, men lost significantly more body weight following the weight loss phase than women (12.8 ± 3.9 vs. 10.1 ± 2.8 kg or 11.7 ± 3.0 vs. 10.6 ± 2.5%, both *p* < 0.001, respectively) (Figure 2A). This difference remained significant after adjustment for age and body weight at baseline (adj. *p* < 0.001). Men regained on average 1.6 ± 5.3 kg following the weight maintenance phase whereas women lost −0.5 ± 5.5 kg. The difference in weight change during the weight maintenance phase remained significant after adjustment for age, weight at baseline and weight lost during the weight loss phase (adj. *p* < 0.001).

### 3.3. Changes in Insulin Sensitivity Following Weight Loss

Several differential changes in glucose homeostasis parameters between men and women were observed following the weight loss phase in the fully adjusted model (see ‘Statistical analysis’ for details). For both HOMA-IR, HIRI and MISI, men significantly improved more following the weight loss phase compared to women (adj. stdβ = 0.104, 0.073 and −0.116, *q* = 0.008, *q* = 0.028 and *q* = 0.015, respectively) (Figure 2D–F). There was a significant decrease in fasting glucose, 2hr glucose and ATIRI in both men and women following LCD-induced weight loss with no differences between sexes (Figure 2B,C,G). During the weight maintenance phase, no significant differences were found between men and women regarding parameters related to glucose homeostasis (Figure 2B–G).

### 3.4. Changes in Lipid Profile and Blood Pressure Following Weight Loss

Following LCD-induced weight loss, men demonstrated a more pronounced decrease in TAG (adj. stdβ = 0.105, *q* < 0.001), total cholesterol (adj. stdβ = 0.083, *q* = 0.030), HDL-cholesterol (adj. stdβ = −0.135, *q* < 0.001), LDL-cholesterol (adj. stdβ = 0.088, *q* = 0.028), adiponectin (adj. stdβ = 0.083, *q* = 0.030) and DBP (adj. stdβ = 0.091, *q* = 0.025), and a greater increase in HDL-cholesterol (adj. stdβ = −0.135, *q* < 0.001) compared to women (Figure 2H,J,K,N,P). In contrast, following the weight maintenance phase, in the fully adjusted model, a greater worsening in HDL-cholesterol (adj. stdβ = 0.164, *q* < 0.001) and a trend for greater worsening in HIRI (adj. stdβ = −0.125, *q* = 0.075), TAG (adj. stdβ = −0.096, *q* = 0.066) and SBP (adj. stdβ = −0.097, *q* = 0.060) were observed in men compared to women (Figure 2E,H,K,O). CRP concentrations decrease more in the weight maintenance phase in men compared to women (adj. stdβ = 0.139, *q* = 0.008). The changes in FFA, total cholesterol, LDL-cholesterol, adiponectin and DBP did not significantly differ between men and women during the weight maintenance phase (Figure 2I–K,P).

### 3.5. Changes in Plasma Lipidome Profile Following Weight Loss

Before the start of the LCD (CID1), men showed significantly higher relative abundance of TAG, DAG and PEO, but lower relative abundance (expressed as percentage of total plasma lipids) of CholE, LPE, PC, PCO, PE and SM than women (Table 2). Furthermore, higher relative abundance of saturated, even and very long TAG species were observed at CID1 in men. Sex-specific changes in the sum scores of plasma lipids were found during the weight loss as well as weight maintenance phase. In the fully adjusted model, men showed a greater decrease in TAG (adj. stdβ = 0.077, *q* = 0.031), CholE (adj. stdβ = 0.113, *q* = 0.008) and SM (adj. stdβ = 0.119, *q* = 0.008) following LCD-induced weight loss compared to women (Figure 3A). In contrast, women demonstrated a greater decrease in LPC (adj. stdβ = −0.162, *q* < 0.001), LPCO (adj. stdβ = −0.140, *q* < 0.001) and a trend for greater decrease in LPE (adj. stdβ = −0.075, *q* = 0.073) following the weight loss phase than men (Figure 3A). For the weight maintenance phase, women showed a less pronounced increase in TAG (adj. stdβ = −0.153, *q* < 0.001), DAG (adj. stdβ = −0.114, *q* = 0.006) and LPC (adj. stdβ = −0.186, *q* < 0.001) compared to men, whilst men showed a smaller increase in SM (adj. stdβ = 0.122, *q* = 0.004) (Figure 3B). Standardized betas for the link between sex and individual lipid species of the lipids within a statistically significant lipid subclass are depicted in Appendix A.

## 4. Discussion

The present study demonstrates sexual dimorphism in LCD-induced body weight loss, improvements in cardiometabolic risk factors, and maintenance of beneficial effects 6 months after weight loss in the large pan-European DiOGenes trial. More specific, men showed greater weight loss and greater improvements in several cardiometabolic risk factors following an 8-week LCD (~800 kcal/day), whereas women demonstrated lower regain of weight, and smaller deterioration of several cardiometabolic risk factors, independent of weight change and diet, during the weight maintenance period 6 months after the weight loss. Interestingly, plasma lipidomic analysis revealed that several lipid species respond in a sex-specific manner to weight loss and weight maintenance, independent from changes in body weight. These findings may have important implications for the development of more targeted dietary interventions to prevent or delay the progression of cardiometabolic complications associated with obesity.

We found more pronounced weight loss following the LCD in men compared to women. One could argue that this may be attributed to a greater energy restriction in men, given the on average greater energy requirements than women. Interestingly, however, men improved more in HOMA-IR, HIRI, MISI, TAG, HDL-, LDL- and total cholesterol, diastolic blood pressure, triglycerides, cholesterol esters, sphingomyelins and adiponectin compared to women, independent of body weight loss. Importantly, the observed differences are also independent of the more beneficial cardiometabolic risk profile at baseline as observed in women compared to men, which is reflected in greater (whole body and liver) insulin sensitivity, higher HDL-cholesterol concentration, lower blood pressure and lower concentration of several lipid species including (saturated and even chain) TAG and DAG. Furthermore, during the 6-month weight maintenance phase, HDL-cholesterol, TAG and DAG had a lower rebound in women, independent of body weight change and diet. These findings suggest that other factors than weight change per se are important determinants of the cardiometabolic changes that occur during weight loss and weight maintenance in men and women.

Based on the present findings, women appear to be less responsive to the dietary intervention with respect to cardiometabolic outcomes. These findings are in line with several previous studies, showing that following weight loss, men lose more weight and improve more in amongst others blood pressure, cholesterol concentration, plasma TAG and HbA1c compared to women [12,13,14,25,26]. It is tempting to postulate that the smaller fluctuations in cardiometabolic risk factors in premenopausal women may be due to sex-differences in metabolic homeostasis and result in a relative protection against metabolic perturbations. Indeed, for a given BMI, women are relatively protected against lipid-induced insulin resistance [27,28] and show higher HDL-cholesterol and lower TAG concentration compared to men [29]. This relative metabolic protection may relate to differences in body fat distribution and ectopic fat storage as a result of differences in substrate supply and utilization, and storage and mobilization of excess lipids, as reviewed elsewhere [4]. Following weight loss, more pronounced changes in intra-abdominal or visceral adipose tissue (VAT) are generally observed in men, whereas women show a more pronounced decrease in (lower body) SAT [30,31,32]. In line, VAT area increased more with weight regain in young overweight men compared to women when expressed as percentage weight regain [33]. VAT mass is positively associated with cardiometabolic risk factors such as insulin resistance and dyslipidemia [34,35], while the expandability of SAT is a critical factor in the prevention of cardiometabolic diseases, and seems to act as an important ‘metabolic lipid sink’ [7,36]. Unfortunately, we have no data on changes in VAT mass available in the current study.

Sexual dimorphism in metabolic homeostasis is also observed in relation to (tissue-specific) insulin sensitivity and glucose homeostasis [4,10]. In the present study, women showed lower whole-body and liver insulin resistance, with similar muscle insulin sensitivity compared to men. Interestingly, women responded less favorable to weight loss with regard to HOMA-IR, HIRI and MISI (independent of baseline insulin sensitivity). It can be speculated that a greater VAT mass loss in men following weight loss, as previously reported, could explain the observed differences in tissue-specific insulin resistance. Insulin signaling in VAT is generally more disturbed compared to SAT [37] and greater VAT mass is linked to the development of liver insulin resistance via FFA release from VAT into the portal vein, affecting glucose and lipid metabolism within the liver [38].

The present plasma lipidomic analyses revealed that LPC decreased more pronounced following weight loss and increased less during weight maintenance in women compared to men. Previously, reduced concentration of LPC have been linked to (muscle) insulin resistance and type 2 diabetes mellitus [10,39]. Moreover, baseline SM concentrations were higher, and SM concentrations decreased less during weight loss and increased more during follow-up in women than men. In contrast to LPC, higher concentrations of SM seem to be related to a more detrimental metabolic health profile including insulin resistance [40,41]. These observations seem to contrast to the more beneficial cardiometabolic profile observed in women. An explanation for this is currently lacking but may possibly reflect a sexual dimorphism in the relationship of these lipidome components and insulin resistance and/or cardiometabolic risk, as previously reported for relationship of TAG and DAG species and hepatic insulin resistance [10]. Interestingly, also diet composition seems to mediate the effect on certain lipid species in both sexes, with the high-protein diets inducing more pronounced increases in PCO and PEO. Further research should elucidate the relationship between LPC and SM and cardiometabolic disease risk in men and women, as well as the impact of diet composition.

To our knowledge, this is the first study to report sex-specific changes in plasma lipid species and tissue-specific insulin resistance following weight loss and during follow-up weight maintenance, independent of the differences in weight change. More detailed measures of body composition are needed to elucidate whether the observed sex-specific changes are at least partly driven by differences in (changes in) body fat distribution. Additionally, sex hormones may affect differences in lipid handling and insulin sensitivity. In the present study, we could not directly explain our results by hormonal status, since most women were in the premenopausal state. Additionally, more subtle effects of sex hormones could not be determined since no data was available on sex hormones or phase of menstrual cycle. Nevertheless, the present findings contribute to a better understanding of the metabolic underpinnings of sex-specific changes following weight loss and weight maintenance.

In conclusion, in this study we observed that overweight or obese men lose more weight and improve more in several cardiometabolic risk parameters following a low-calorie diet but are less able to maintain several of the improvements after 6 months of follow up compared to premenopausal women. Furthermore, several plasma lipid species changed in a sex-specific manner, independently of the direction of the weight loss, which may have implications for the sex-specific development of cardiometabolic diseases. Altogether, these findings may provide directions for more sex-targeted dietary interventions in the prevention of cardiometabolic diseases.

## Figures and Tables

**Figure 1 nutrients-13-01588-f001:**
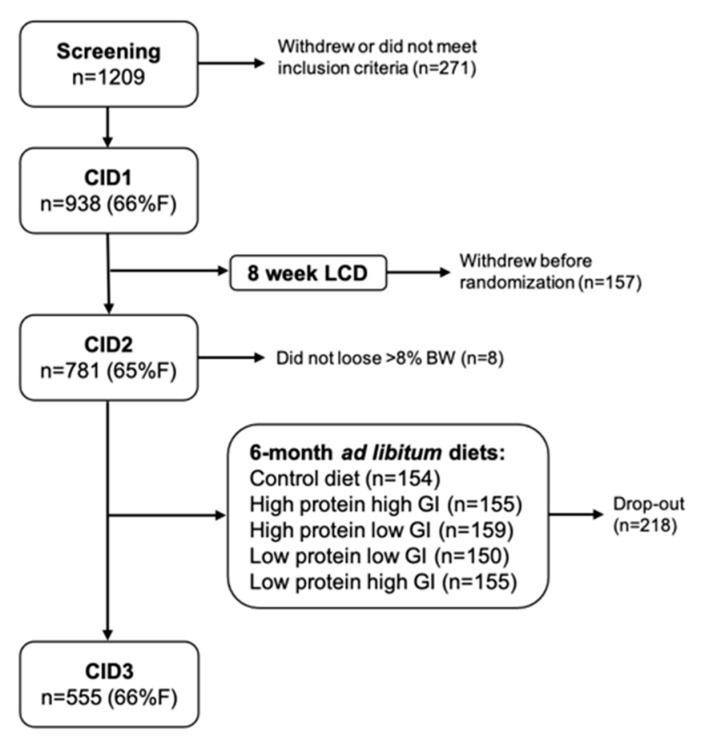
Schematic overview of participant inclusion and study design of the DiOGenes study. %F = percentage of females.

**Figure 2 nutrients-13-01588-f002:**
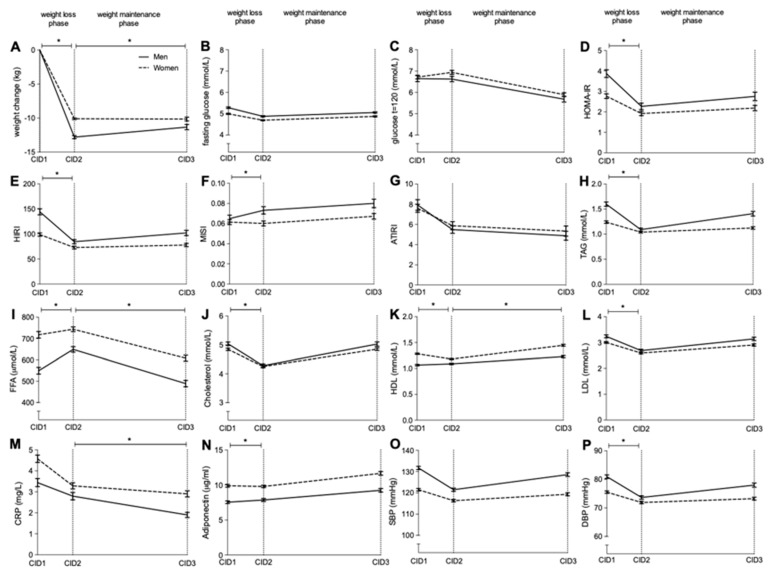
Mean metabolic changes in men and women following an 8-week weight loss phase (from CID1 to CID2) and after 6 months of weight maintenance (CID2 to CID3) ± standard error. Change in weight (**A**), fasting glucose (**B**), glucose t = 120, |(**C**), HOMA-IR (**D**), HIRI (**E**), MISI (**F**), ATIRI (**G**), TAG (**H**), FFA (**I**), cholesterol (**J**), HDL (**K**), LDL (**L**), CRP (**M**), adiponectin (**N**), SBP (**O**) and DBP(**P**) are presented.The black line represents men and the dotted line represents women. * Indicates a significant difference in change between men and women (*q* < 0.05) in a linear mixed model corrected for age, body weight (changes) and baseline differences.

**Figure 3 nutrients-13-01588-f003:**
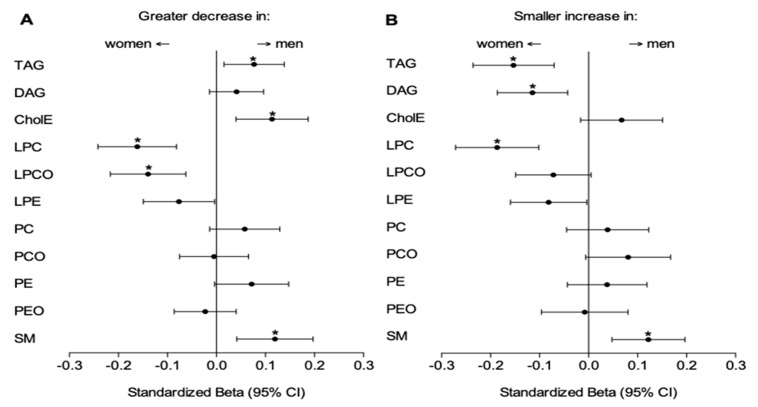
Standardized Beta’s ± 95% confidence interval (CI) for the association between sex and the change in lipid groups following weight loss (from CID1 to CID2) (**A**) and after 6 months of weight maintenance (from CID2 to CID3) (**B**). A negative Standardized Beta indicates a greater decrease (**A**) or smaller increase in women (**B**) compared to men and vice versa. * Indicates a significant difference in change between men and women (*q* < 0.05 (=False Discovery Rate *p*-Value)) in a linear mixed model corrected for age, body weight (changes) and baseline differences.

**Table 1 nutrients-13-01588-t001:** Participant characteristics.

	n	Men	Women	Total Group	*p*-Value
Sex (%)	782	35.3	64.7 *	100	<0.001
Age (years)	782	42.5 ± 6.0	41.0 ± 6.3 *	41.5 ± 6.26	0.001
BMI (kg/m2)	782	34.3 ± 0.5	34.5 ± 5.0	34.4 ± 4.9	0.108
Body weight (kg)					
CID1	782	109.3 ± 17.4	95.0 ± 15.6 *	100.1 ± 17.6	<0.001
Weight loss CID2-CID1	782	−12.8 ± 3.9	−10.1 ± 2.8 *	−11.0 ± 3.50	<0.001
Weight maintenance CID3-CID2	555	1.6 ± 5.3	−0.5 ± 5.5 *	0.5 ± 5.48	<0.001
Physical activity (Baecke)	718	7.9 ± 0.9	8.0 ± 0.9	8.0 ± 0.9	0.090
Waist:hip ratio	769	1.01 ± 0.05	0.88 ± 0.07 *	0.92 ± 0.09	<0.001
Systolic BP (mmHg)	758	132.0 ± 12.8	121.9 ± 14.4 *	118.2 ± 13.3	<0.001
Diastolic BP (mmHg)	758	81.4 ± 10.6	75.9 ± 10.6 *	72.49 ± 9.72	<0.001
Fasting glucose (mmol/L)	749	5.27 ± 0.62	5.00 ± 0.66 *	5.10 ± 0.66	<0.001
2 hr glucose (mmol/L)	741	6.58 ± 2.37	6.74 ± 2.02	6.68 ± 2.16	0.333
Fasting insulin (μIU/mL)	723	13.9 ± 11.3	10.5 ± 9.4 *	11.7 ± 10.3	<0.001
HOMA-IR (A.U.)	733	3.87 ± 3.23	2.77 ± 2.64 *	3.16 ± 2.91	<0.001
MISI (A.U.)	649	0.063 ± 0.052	0.060 ± 0.052	0.061 ± 0.052	0.488
HIRI (A.U.)	686	144.8 ± 101.8	98.9 ± 76.0 *	115.9 ± 89.1	<0.001
ATIRI (A.U.)	720	7.92 ± 8.95	7.69 ± 8.38	7.72 ± 8.61	0.625
Total cholesterol (mmol/L)	768	5.05 ± 1.08	4.84 ± 0.96 *	4.92 ± 1.01	0.008
HDL cholesterol (mmol/L)	770	1.07 ± 0.28	1.29 ± 0.33 *	1.21 ± 0.33	<0.001
LDL cholesterol (mmol/L)	764	3.24 ± 0.93	3.00 ± 0.84 *	3.08 ± 0.88	<0.001
TAG (mmol/L)	759	1.60 ± 0.70	1.23 ± 0.57 *	1.36 ± 0.64	<0.001
FFA (μmol/L)	672	552.3 ± 281.6	711.54 ± 326.7 *	646.8 ± 318.7	<0.001
C-reactive protein (mg/L)	747	3.78 ± 3.18	4.54 ± 4.13 *	4.12 ± 3.86	<0.001
Adiponectin (μg/mL)	768	7.46 ± 3.17	9.96 ± 4.71 *	9.08 ± 4.40	<0.001

Values are represented as mean ± standard deviation (SD) unless otherwise indicated. * Significant difference compared to men (*p* < 0.05) CID: clinical investigation day, CID1: baseline measurements, CID2: measurements following the weight loss phase, CID3 measurements following the weight maintenance, BP = blood pressure, HOMA-IR = homeostatic model assessment for insulin resistance, MISI = muscle insulin sensitivity index, HIRI = hepatic insulin resistance index, ATIRI = adipose tissue insulin resistance index, TAG = triacylglycerol, FFA = free fatty acid.

**Table 2 nutrients-13-01588-t002:** Relative abundance individual lipid groups at baseline (CID1).

	Men (n = 258)	Women (n = 474)	Total Group (n = 732)	*p*-Value
TAG	50.9 ± 8.2	44.8 ± 8.9 *	47.0 ± 9.2	<0.001
Odd	5.7 ± 1.1	6.4 ± 1.3 *	6.2 ± 1.3	<0.001
even	94.3 ± 1.1	93.6 ± 1.3 *	93.8 ± 1.3	<0.001
Sat	2.5 ± 1.1	2.3 ± 1.0 *	2.4 ± 1.07	0.022
Usat	76.0 ± 5.2	76.2 ± 5.6	76.1 ± 5.4	0.649
Psat	21.5 ± 5.7	21.5 ± 6.0	21.5 ± 5.9	0.992
Long	95.6 ± 1.2	95.9 ± 1.3 *	95.8 ± 1.3	0.004
very long	4.4 ± 1.2	4.1 ± 1.3 *	4.2 ± 1.3	0.004
DAG	0.040 ± 0.014	0.037 ± 0.015 *	0.038 ± 0.015	0.047
CholE	0.13 ± 0.04	0.15 ± 0.04 *	0.14 ± 0.04	<0.001
LPC	5.7 ± 1.5	5.8 ± 1.8	5.8 ± 1.7	0.758
LPCO	0.035 ± 0.011	0.037 ± 0.014	0.036 ± 0.013	0.098
LPE	0.033 ± 0.014	0.037 ± 0.018 *	0.036 ± 0.013	0.003
PC	34.1 ± 5.4	38.6 ± 5.9 *	37.0 ± 6.1	<0.001
PCO	1.33 ± 0.42	1.45 ± 0.43 *	1.40 ± 0.43	<0.001
PE	1.73 ± 0.43	2.08 ± 0.50 *	1.95 ± 0.51	<0.001
PEO	0.37 ± 0.15	0.34 ± 0.12 *	0.35 ± 0.13	0.047
SM	5.6 ± 1.4	6.6 ± 1.7 *	6.3 ± 1.6	<0.001

Lipid groups are represented as percentage of all lipids ± standard deviation (SD) except for TAG subgroups who are expressed as percentage of total TAG ± SD. * Significant difference compared to men (*p* < 0.05). TAG = triacylglycerol (n = 55), TAGodd = with an odd number of carbon atoms, TAGeven = with an even number of carbon atoms, TAGsat = without double bonds, TAGusat = with 1–3 double bonds, TAGpsat = with ≥4 double bonds, TAGlong = with <56 carbon atoms, TAGverylong = with ≥56 carbon atoms, DAG = diacylglycerols (n = 2), CholE = cholesterol esters (n = 3), LPC = lysophosphatidylcholine (n = 11), LPCO = lysoalkylphosphatidylcholine (n = 2), LPE = lysophosphatidylethanolamine (n = 1), PC = phosphatidylcholine (n = 25), PCO = alkyl-phosphatidylcholine (n = 15), PE = phosphatidylethanolamine (n = 3), PEO = alkylphosphatidylethanolamine (n = 3) and SM = sphingomyelins (n = 20).

## Data Availability

Data described in the manuscript will be made available upon request pending.

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
