# Peer review of "Sexual Dimorphism in Body Weight Loss, Improvements in Cardiometabolic Risk Factors and Maintenance of Beneficial Effects 6 Months after a Low-Calorie Diet: Results from the Randomized Controlled DiOGenes Trial"

_nutrients, 2021, doi:10.3390/nu13051588_

Round 1
Reviewer 1 Report
Authors describe an in depth analysis of the lipidome on a relatively large cohort of men and women following weight-loss and a weight maintenance program and illustrate some sexual differences in response to these interventions.
Statistical methods used are appropriate and the manuscript is clearly written and presented.
I only have a couple of very minor questions / comments;
The females in the study were pre and post menopausal. With the premise that female hormone profile might contribute to the findings, it would help if the proportion of females who were post-men' was indicated.
Diet was used as a covariate in the analysis. Did authors see any moderating effect of diet on the lipid variables?
Only spotted a couple of typos on the manuscript (line 48 unnecessary use of 'in women'), and an 's' missing on 'species' in the supplementary table description.
Reviewer 2 Report
It is an interesting article, and intensive work on a large cohort, that presents several advantages as it demonstrated the gender differed lipid profiles on diet experiments. In this article, the authors have evaluated cardiometabolic risk factors in men and women following weight loss and after weight maintenance. As evidenced by numerous articles, even modest weight reduction in obese and overweight individuals can reduce the risk factors for diabetes and cardiovascular disease (CVD). I have only one major concern about the coverage of lipid species. In general, the plasma lipidome covers 400-500 lipid species, but here in this article authors have identified only 140 lipid species. Is it might be putative identification? Authors can briefly describe the LC-MS methodology and feature identification and confirmation instead of just referring to previous articles.
